# Production, Composition, and Ecological Function of Sweet-Basil-Seed Mucilage during Hydration

**Dongfang Zhou** , **Jacob N. Barney** and **Gregory E. Welbaum** *

School of Plant and Environmental Sciences, Virginia Tech, Room 301C Saunders Hall, West Campus Drive, Blacksburg, VA 24061, USA; dfezhou@vt.edu (D.Z.); jnbarney@vt.edu (J.N.B.)
* Correspondence: welbaum@vt.edu; Tel.: +1-540-231-5801; Fax: +1-540-231-3083

**Abstract:** The sweet-basil (*Ocimum basilicum* L.) fruit/pericarp produces mucilage that engulfs the fruit and seed within minutes of hydration. Seed mucilage is produced by plant species that have adapted to arid, sandy soils. This study was conducted to determine how basil-seed mucilage improves ecological fitness. A second objective was to find ways to remove mucilage, which may interfere with commercial planting. Basil fruit/seeds were examined using light and environmental scanning electron microscopy. Columnar structures of basil mucilage rapidly unfolded from the pericarp upon initial hydration. Dilute hydrochloric acid removed the mucilage, which decreased the water content four-fold but did not inhibit seed germination in a laboratory test. Nondestructive Fourier-transform mid-infrared (FTIR) spectroscopy confirmed that the mucilage was primarily composed of hemicellulose that anchored the basil seed to resist movement. The fully hydrated seeds approached zero water potential, so the mucilage did not interfere with hydration. The seeds that were planted in growing media with mucilage had from 12 to 28% higher seedling emergence and survival percentages after 10 days than seeds without mucilage. Basil-fruit/seed mucilage provides a reservoir of loosely bound water at high water potential for seed germination and early seedling development, thus improving survivability under low moisture.

**Keywords:** seed biology; seed ecology; stand establishment; mucilage removal; germination; myxodiaspory; seed–water relations

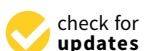



## 1. Introduction

Mucilage is produced by many diverse plant species and different tissues including seeds, leaves, and roots [1]. Mucilage likely has various ecological functions such as water storage, especially in succulent species [2]. Mucilage may lubricate growing roots as they penetrate soil [3] and may assist carnivorous plants in trapping prey [4].

Double fertilization in angiosperms leads not only to embryogenesis and development of the endosperm, but also to the differentiation of surrounding ovule integuments that become the seed coat. The seed coat in some species has evolved special functions to protect the embryo, facilitate dispersal, and support germination. One specialization is the deposition of hydrophilic mucilage in the outer layers of certain seed structures that aids in dispersal [5]. The production of hydrophilic mucilage by the seed coat or fruit pericarp during hydration, known as myxodiaspory, is common in the angiosperm families Acanthaceae, Asteraceae, Brassicaceae, Lamiaceae, Liaceae, Plantaginaceae, and Poaceae [6–10]. Mucilage composition varies among species but is primarily composed of hydrophylic pectins and hemicelluloses that undergo substantive swelling when hydrated. Sweet-basil (*Ocimum basilicum* L.)-fruit mucilage is similar to *Arabidopsis*, flax (*Linum usitatissimum*) and chia (*Salvia hispanica*), but contains arabinoxylan and glucomannan [11–13]. Basil-seed polysaccharides extracted in cold water with alcohol precipitation are reportedly composed of two major fractions: (i) an acid-stable core glucomannan (43%) having a ratio of glucose to mannose of 10:2, and (ii) a (1→4)-linked xylan (24.3%) having acidic side chains at C-2 and C-3 of xylosyl residues in the

acid-soluble fraction. Additionally, a minor glucan fraction (2.3%) from cellulose degradation was also reported [13].

Mucilage, produced by either the seed coat or pericarp, plays ecophysiological roles including facilitation of seed hydration, regulation of germination by affecting oxygen entry into seeds, improved germination in saline environments, and mediation of seed dispersal through adhesion to soil or animal vectors [6,9,10,14–16]. For example, the achenes of *Artemisia sphaerocephala* (Asteraceae) produced mucilage and germinated to a higher percentage compared with demucilaged achenes [17].

In addition to seed ecology, basil-seed mucilage is consumed as a health food. Hydrated basil seeds with mucilage are bottled as a popular drink in parts of the world. Sweet-basil seeds are also consumed to reduce craving for food because the mucilage tends to fill the stomach. Therefore, basil seeds with mucilage are of interest as a health food with possible medicinal properties.

Because mucilage can interfere with planting equipment under high humidity, we developed treatments to remove it without interfering with germinability and vigor. To better understand the role of mucilage in basil reproductive ecology, seeds with and without mucilage were germinated on different substrates and surfaces. The water relations of basil-seed hydration were studied to understand how mucilage may aid in establishment. Mucilage was analyzed via Fourier-transform mid-infrared (FTIR) spectroscopy, a novel nondestructive rapid technique, to confirm the chemical composition reported in earlier studies.

## 2. Materials and Methods

### 2.1. Plant Material

Three distinct types of commercial sweet-basil cultivars were selected for study: 'Italian Large Leaf', 'Genovese', and 'Aroma 2' (Johnny's Seed Company, Fairfield, ME, USA). 'Italian Large Leaf' is a higher-yielding basil cultivar, with a sweeter and less clove-like scent and taste compared with 'Genovese', which is tender, fragrant, with extra-large, dark-green leaves. 'Aroma 2' is an F1 hybrid, fusarium wilt-resistant cultivar adapted to greenhouse or field production. All had >87% germination over the temperature range of 18–35 °C [18].

### 2.2. Effect of Mucilage on Seed Moisture Content

Pericarp mucilage was removed both chemically and mechanically from basil seeds. Gentle hand scarification with fine sandpaper mechanically removed the mucilage-containing pericarp layer from dried seeds. For chemical removal, concentrated hydrochloric acid (HCl, 37.7% Fisher Scientific, Pittsburgh, PA, USA) was diluted with distilled water at a ratio of 1:2 *v/v* to yield 12.6% HCl. Dry basil seed (250 mg) was added to 50 mL of dilute acid solution, stirred for 5 min., filtered through a 1 mm sieve, washed with running water until all visible mucilage was removed, and dried at 25 °C at 40% RH. The moisture content of basil seeds with and without mucilage was measured after 1 h hydration in distilled water on germination blotter paper (Anchor Paper, St. Paul, MN, USA). Seeds were oven dried for 16 h at 103 °C and cooled in a desiccator at 22 °C before weighing [19]. Moisture content values are means ± SE of 25 seeds.

### 2.3. Mucilage Visualization

Thin sectioning for visualization was as follows: (1) Seeds were placed in 10% formalin, (2) dehydrated in a tissue processor (Leica EM, Buffalo Grove, IL, USA) in formalin, 70% alcohol, 80% alcohol, 3 changes of 95% alcohol, 3 changes of 100 % alcohol, 2 changes of xylene, and 4 changes of liquid paraffin wax. Tissue-processor steps included 1 h of duration for all solutions except paraffin, which was 30 min. each. (3) Dehydrated seed samples were imbedded into paraffin and (4) cut with a rotary microtome into 4-micron sections. The thin sections were floated on water and picked up on microscope slides. Slides were dried at 60 °C for 20 min. with forced air to melt paraffin so that the tissue would adhere to each slide and remove excess water. (5) Paraffin was removed in 2 changes

of xylene for 3 min. each. Slides were next rehydrated in 95% alcohol, 2 changes for 3 min. each, and then rinsed in running tap water in an automated stainer. (6) Slides were next rinsed in distilled water and (7) stained in 0.002% toluidine blue for 4 min. and dehydrated sequentially in 95%, 100% alcohol, and 95% xylene. (8) Samples were protected by cover slips attached with resinous mounting media.

Hydration of unprocessed basil seeds were visualized and recorded with a DinoXcope (Dino-Lite Digital Microscope, Torrance, CA, USA) connected to a desktop computer. Hydrated seeds were treated with 0.1% toluidine blue, which stains nucleic acids blue and polysaccharides purple while increasing image sharpness. Thin processed sections were visualized by light microscopy (Olympus SZX16). The surface structure of dry intact seeds, demucilaged seeds, and dead seeds that produced no mucilage were visualized by ESEM (Environmental Scanning Electronic Microscopy) (Manufacture: FEI, Instrument: QNANTA 600F). Dry intact, demucilaged and dead seeds were sputter coated with Au/Pd (10 nm thickness; Cressington Sputter Coater, 208HR) to increase image resolution prior to ESEM.

### 2.4. Mucilage Analysis Using FTIR

Twenty 'Italian Large Leaf' basil seeds were hydrated for 1 h in distilled water. Individual seeds were held with dissecting forceps (Fisher Scientific, Pittsburgh, PA, USA), and mucilage from around the seed was carefully excised with a scalpel into a plastic container. The mucilage was dried at 30% RH in a desiccator. Dried mucilage residue was collected in 5 mL microfuge tubes, sealed, and stored in a refrigerator until analyzed.

Rapid non-destructive biochemical analysis of mucilage was performed by FTIR spectroscopy at the mid-infrared beamline (01B1-1) of the Canadian Light Source (Saskatoon, SK, Canada). The dried mucilage was ground to a powder using Genogrinder 201 (SPEX SamplePrep, LLC, Metuchen, NJ, USA). The powdered mucilage (~1.0–1.2%) was mixed with KBr and pressed into 13 mm diameter pellets with an automatic press (AutoCrushIR PIKE Technologies Inc., Madison, WI, USA). Three pellets of mucilage were analyzed by FTIR in the transmission mode from 800–4000 $cm^{-1}$ at a spectral resolution of 2 $cm^{-1}$ using the Agilent FTIR spectrometer (Cary 670 series, Agilent Technologies Inc., Santa Clara, CA, USA) with a glowbar source. The spectra of mucilage were compared with reference plant biopolymer spectra for: cellulose, hemicelluloses (xylan from spelt oats: hot-water-insoluble xylan, hot-water-soluble xylan, and wheat arabinoxylan), apple pectin and oat β-glucan. The reference compounds were purchased either from Sigma-Aldrich Ltd. (Oakville, ON, Canada) or Megazyme International Ltd. (Lansing, MI, USA). Each sample spectrum was an average of 64 scans. A pure KBr spectrum with 128 scans was recorded as a background for correcting all spectra.

The FTIR spectra were then normalized by sample weight, Gaussian smoothed by 3 points, and baseline corrected using Rubber Band Correction and Orange software (version 3.13, Bruker Optik, Ettlingen, Germany) [20]. Second derivatives (after which the spectra were smoothed by 20 points) were plotted from the spectra with OriginPro 2018 (OriginLab Corporation, Northampton, MA, USA) software.

### 2.5. Seed Hydration and Dehydration Time Courses

'Italian Large Leaf' and 'Aroma 2' seeds were chosen to compare hydration with and without mucilage because these two cultivars had greater than 90% germination over a range of 18–35 °C [18]. Replications of 50 seeds were hydrated on three thicknesses of 47 mm-diameter, fully hydrated (4.5 mL) filter papers (Whatman, Qualitative 1, Fisher, Pittsburgh, PA, USA) in small Petri plates (aseptic 47 mm, Fisher Scientific, Pittsburgh, PA, USA) at 22 °C. The top layer of filter paper containing seeds was removed with forceps every 15 min. and weighed for the first hour, then at 30 min. intervals until the first radicle emerged or until weight stabilized. Seed weight was determined by subtracting the weight of the hydrated filter-paper sheet and averaged across replications.

Fully hydrated seeds, as described above, from each replicate were transferred to a small 5 cm-diameter aluminum dish and dried in a 0.5 m$^3$ drying oven (Fisher Scientific, Pittsburgh, PA, USA) at 103 °C for >16 h [19]. Water content (WC) was calculated as:

$$\text{WC} = \frac{\text{Fresh Weight (FW)} - \text{Dry weight (DW)}}{\text{Dry Weight (DW)}}$$

*2.6. Sweet-Basil-Mucilage Osmotic Pressure*

Germination of 'Italian Large Leaf' basil seeds was tested by equidistant placement on two thicknesses of germination blue blotter paper (Anchor Paper Co., St. Paul, MN, USA) inside 110 × 110 × 35 mm sealed clear acrylic boxes (Hoffman Manufacturing, Inc., Corvallis, OR, USA) moistened to saturation with 15 mL distilled water or an osmotic solution of polyethylene glycol (PEG) 8000 molecular weight (MW) (MilliporeSigma, Burlington, MA, USA). PEG was used as a nonpenetrating, nonphytotoxic osmoticum. Seeds were hydrated in PEG solutions of −0.25 and −0.5 megapascals (MPa), verified by osmometry (5500 Vapor Pressure Osmometer, Wescor, Logan, UT, USA) at an experimental temperature of 25 °C. Bottoms of acrylic germination boxes were smoothed by hand filing to ensure continuous contact with the surface of the one-dimensional, thermogradient table (Thermogradient Systems LLC, Blacksburg, VA, USA). Boxes were randomized within each temperature in the dark on the gradient table.

Osmotic potential of 'Italian Large Leaf' and 'Aroma 2' seed mucilage was measured by osmometry after excision from 20 fully hydrated seeds in a 1 mL osmometer sample chamber. The osmometer was operated in the manual cooling mode to extended equilibration times. Each sample was read twice, and values varied less than 5 mOsmols/kg. The osmolality readings from vapor pressure osmometer were converted to water potential (Ψ) using Van't Hoff equation [21].

*2.7. Seed Adhesion Tests*

The adhesion of seeds to a flat glass plate tested whether the pericarp mucilage could help anchor seeds. Twenty fully hydrated 'Italian Large Leaf' basil seeds with and without mucilage were placed on a glass plate, and the tilt angle was adjusted until the seeds moved in response to gravity at 25 °C and 39% RH.

*2.8. Germination of Seeds in Media with Different Amounts of Water Added*

Basil seeds were planted in cell trays on growing media (Sunshine # 1, a mixture of coarsely ground, sphagnum moss and perlite, Sun Gro Horticulture, Agawam, MA, USA) of uniform moisture content taken from a previously unopen bag. Black plastic transplant trays, (48 cells, 4 cm (length) 3 cm (width) 5.5 cm (depth) each, 60 cm$^3$ per cell), were equally filled with media. Five seeds of each treatment were placed on top of the media in the corners and one in the middle of 12 transplant tray cells. After planting, 2, 4, or 6 mL of water were pipetted into each cell. The trays were covered with a clear plastic cover and placed in an incubator at 25 °C. No additional water was added. Germination and survival percentage of basil seeds with/without (*w/o*) mucilage was recorded ± SE after 10 d at 25 °C (24 replications of 5 seeds each). Germination was measured by radicle emergence. Surviving seedlings had green cotyledons and white roots after 10 day.

## 3. Results and Discussion

The basil seed is botanically a fruit with a thin outer pericarp layer [22]. Light-microscopy images show the detailed structure of pericarp tissue surrounding the seed coat (Figure 1).

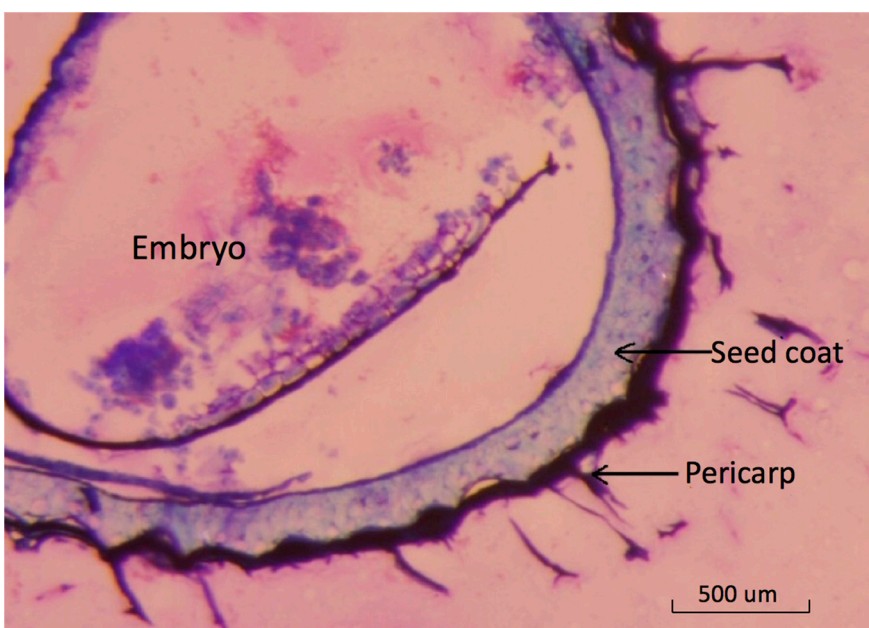

**Figure 1.** Cross section of dry basil fruit showing relationships between embryo seed coat and pericarp tissues. Light-microscopy image of a thin cross section of dried basil fruit stained with toluidine blue showing outer pericarp layer where mucilage originates.

The fruit, with a seed structure inside, is small and black with a shallow undulated surface (Figure 2A). The mean dimensions of the 'Genovese' fruit length, width, and thickness were 2.4 ± 0.2, 1.4 ± 0.3, and 1.1 ± 0.3 mm, respectively. Dry fruit of 'Italian Large Leaf' and 'Aroma 2' had similar exterior appearances and dimensions (not shown).

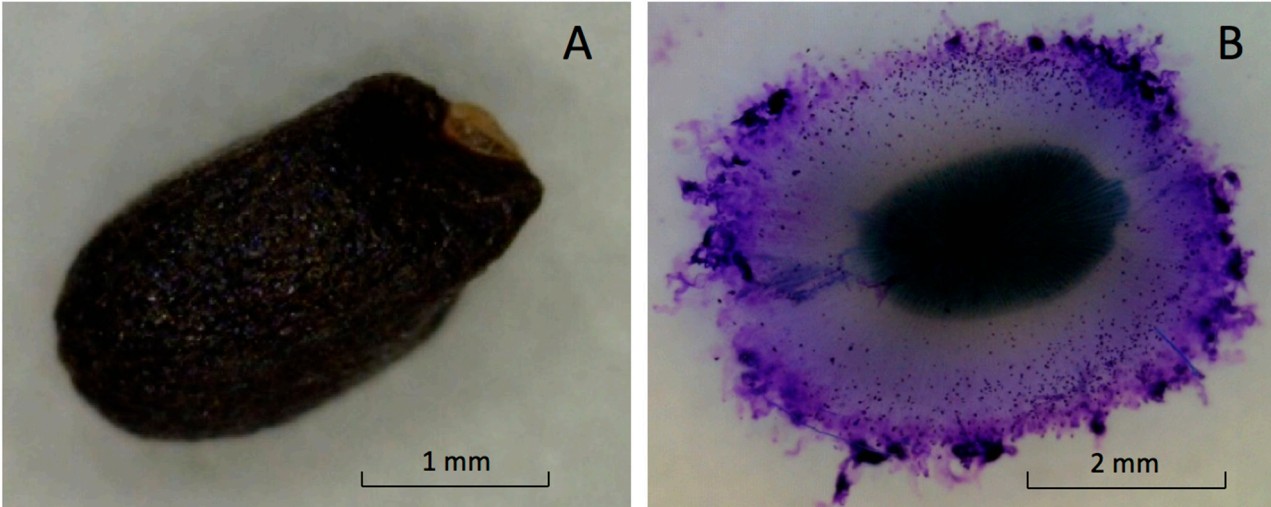

**Figure 2.** Images of intact dry basil seed (**A**) and fully hydrated seed (**B**) with mucilage attached.

Basil is classified in the family Lamiaceae. Some members are myxocarpic and exude polysaccharide upon hydration from fruit the pericarp surrounding the seed [5]. Mucilage excretion increased the total volume of hydrated basil seeds five-fold compared to dry seeds (Figure 2A,B).

Mucilage production began within seconds after imbibition (see Video S1). Within one minute, the seed was covered with a thick layer of mucilage (Figure 3A). This rapid expansion continued until the mucilage layer was fully formed (Figure 3B). The mucilage was tethered to the pericarp by long, differentially stained strands (Figure 3) [5,13]. The outer edge of the mucilage strands was stained bright blue (Figure 3). Almost 90% of

the mucilage was produced within 20 min. of hydration (Figure 3C). Mucilage expanded more slowly after 20 min. and stopped expanding after an hour (Figure 3D). The staining was heterogenous, suggesting the mucilage and strands were composed of different polysaccharides or that the stain would not fully penetrate the mucilage (Figure 3).

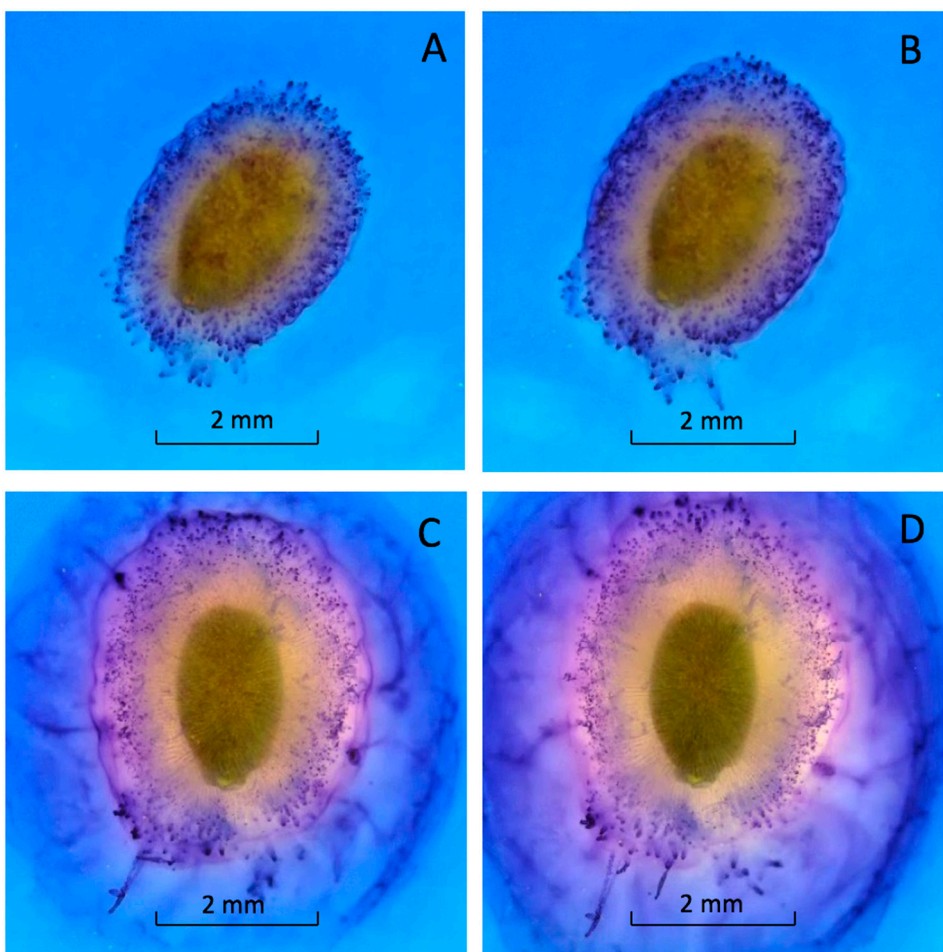

**Figure 3.** Basil-seed-coat mucilage after imbibition in 0.1% toluidine-blue solution. (**A**) 1 min; (**B**) 5 min; (**C**) 20 min; (**D**) 1 h after imbibition. View a video of basil seed hydration (see Video S1).

The structure, bioactive properties, and genetics of mucilage formation have been reviewed from a food-science perspective [23]. Fewer studies have focused on effects of mucilage on the reproductive ecology of plants. Similar to basil, mucilage formation in the apoplast of epidermal cells during pericarp development has been reported in other species as well. Upon hydration, these mucilage deposits quickly form a gelatinous coating around the fruit [5,22]. The role of this rapidly forming mucilage layer has been debated. Seed/fruit mucilage has been proposed to facilitate seed hydration, regulate germination by affecting oxygen entry into seeds, and mediate seed dispersal through adhesion to soil or animal vectors [6,9,10,14,15]. The intact achenes of *Artemisia sphaerocephala* (Asteraceae), which also produce mucilage, have a higher germination percentage than demucilaged achenes [17]. Chia (*Salvia hispanica* L.)-seed mucilage improves seed germination in saline soils [16].

The qualitative FTIR analysis of 'Italian Large Leaf' mucilage produced spectra that were identified through comparison with other biopolymer standards (Figure 4). FTIR was selected to confirm earlier reports of mucilage composition because it is fast, nondestructive, and offers a rapid alternative to HPLC analysis. Direct photochemical qualitative FTIR is free of artifacts caused by hydrolysis and purification. Using HPLC, basil-seed polysaccharides were previously reported to consist of: (i) an acid-stable core glucomannan (43%)

having a ratio of glucose to mannose 10:2, and (ii) a (1→4)-linked xylan (24.3%) having acidic side chains at C-2 and C-3 of xylosyl residues in the acid-soluble portion [13]. In the current study, cellulose and hemicellulose produced distinct peaks that were both part of the 'Italian Large Leaf' mucilage spectrum (Figure 4). The soluble and insoluble xylans are differentiated using the 1637 cm$^{-1}$ and 1603 cm$^{-1}$ peaks, respectively (Figure 4) [24]. The second derivatives of basil-mucilage spectra show similarity to soluble xylan with prominent peaks at 1600 and 1165 cm$^{-1}$ (Figure 4) [24]. Further, the mucilage spectrum has a small peak at 1737 cm$^{-1}$ and prominent peaks at 1454 and 1058 cm$^{-1}$ (Figure 4) that may arise from low or high MW methylated pectin [25]. Basil mucilage is not pure hemicellulose as some earlier reports indicated, since our analysis shows that pectin and cellulose are also present (Figure 4). Except for pectin, our results are consistent with earlier mucilage analyses that used traditional GC and HPLC techniques. This illustrates that FTIR spectrometry provides an alternative for qualitative analysis.

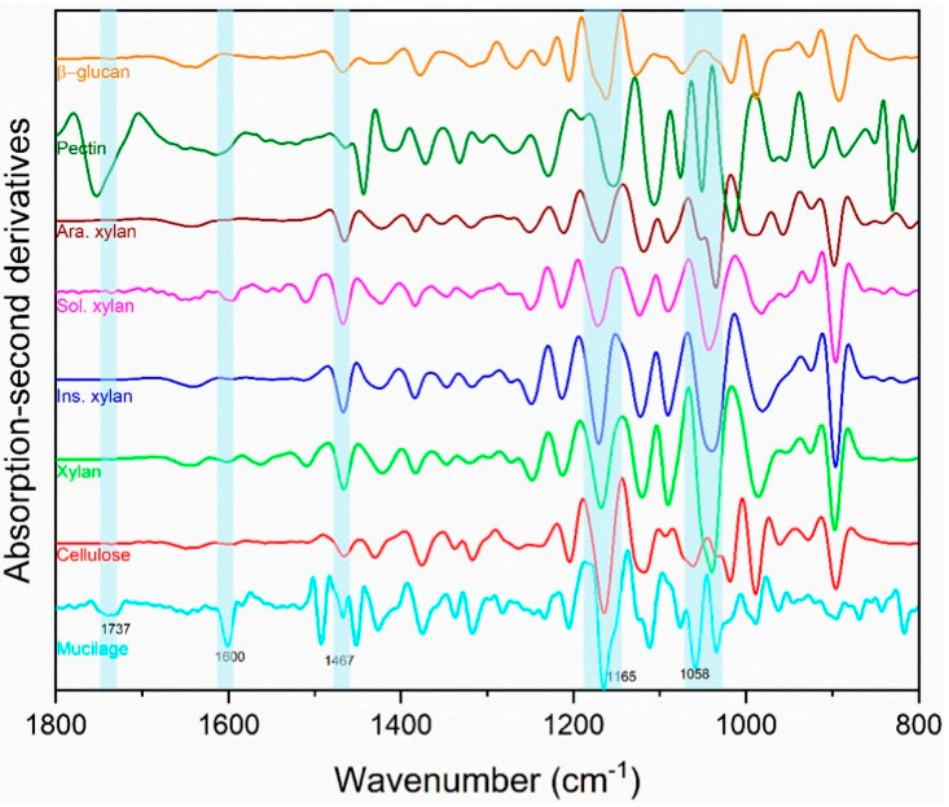

**Figure 4.** Basil-seed-mucilage chemical composition determined nondestructively by Fourier-transform mid-infrared (FTIR) spectroscopy. Second derivatives of FTIR absorption spectra of basil-seed mucilage (mucilage), cellulose, xylan, insoluble xylan, (Ins. xylan), soluble xylan (Sol. xylan), arabinoxylan (Ara. xylan), apple pectin (pectin), and β-glucan.

Basil is like other mucilage-producing seeds such as Arabidopsis [11–13]. In Arabidopsis seed mucilage, cellulosic rays form a scaffold to hold two mucilage domains [26]. Dark staining tips of similar rays are visible in basil-seed images. These columns radiate from the testa, extending through viscous mucilage layer, holding it in place around the seed (Figure 3). Xylan chains mediate the adsorption of mucilage to cellulose microfibrils in Arabidopsis [26]. Xylose and xylan binding to the Arabidopsis cellulose scaffold provide support to the mucilage layer [25]. These polysaccharides are also present in basil and along with pectin give mucilage its thick viscosity (Figure 4).

When planted under wet or humid conditions, basil seed can produce mucilage prematurely, interfering with precision-seeding equipment. Therefore, removing mucilage prior to planting is of interest. The mechanical removal of seed mucilage by rubbing seeds with sandpaper or paper towels was effective but slow, tedious, and impractical for large quan-

tities of seed. Mucilage is commercially removed from wet-seeded crops such as tomato and melon post-harvest by fermentation, acid or base treatment, or enzymatically [27,28]. Sodium carbonate was investigated but did not effectively remove basil mucilage (data not shown).

Diluted hydrochloric acid (HCl) is often commercially favored to remove pubescence, mucilage, and control certain seed-borne diseases. HCl also produces clean, attractive seeds that are easy to singulate. The specific dilution and duration of treatment used by seed companies varies. One popular commercial recipe for removing tomato-seed mucilage involves adding 567 mL of concentrated HCl to 10 L of tomato seed/fruit slurry mixture for 30 min (verbal communication). HCl was diluted (1:2, acid:distilled water) to effectively remove mucilage from 'Italian Large Leaf', 'Genovese' and 'Aroma 2' seeds without significantly reducing germinability (Table 1). Mucilage was removed by HCl in all subsequent experiments.

**Table 1.** Effect of mucilage removal using diluted HCl on germination of 'Genovese', 'Aroma 2' and 'Italian Large Leaf' sweet basil at 32 °C in an incubator on germination blotter paper.

| Genotype and Treatment | Germination (%) † | MTG (Days) ‡ |
|---|---|---|
| 'Genovese' | | |
| 　　intact | 87 ± 1.5 | 1.20 ± 0.08 |
| 　　mucilage removed | 88 ± 2.0 | 1.54 ± 0.07 |
| 'Aroma 2' | | |
| 　　intact | 95 ± 1.0 | 1.22 ± 0.05 |
| 　　mucilage removed | 93 ± 1.5 | 1.65 ± 0.08 |
| 'Italian Large Leaf' | | |
| 　　intact | 96 ± 1.4 | 1.49 ± 0.11 |
| 　　mucilage removed | 94 ± 1.0 | 1.60 ± 0.11 |
| | NS | NS |

† Means of four replications of 25 seeds each for intact type, and three replications of 50 seeds each for those *w/o* (without) mucilage are shown ± SE. ‡ Mean time to germination: $\sum(N_iT_i)/\sum(N_i)$, where $N_i$ is the number of new germinated seeds at time $T_i$ after imbibition, calculated from four replications of 25 seeds each for intact seeds, and three replications of 50 seeds each for demucilaged seeds are shown ± SE. NS is nonsignificant according to paired T-test.

There were no significant differences in germination percentage or the mean time to germination (MTG) between seeds with or without mucilage when Genovese', 'Aroma 2' and 'Italian Large Leaf' were compared (Table 1). The germination percentage was unchanged without the mucilage layer while the MTG increased slightly by half a day. This suggests that the mucilage removal did not affect the seed-germination percentage, but mucilage helped seeds germinate faster in some cases. Acid treatment may have damaged the seeds in a way that did not reduce germination percentage but slowed germination. Seed mucilage may aid in seed hydration by increasing contact with the media, thereby and increasing hydraulic conductivity between the seed and the surrounding environment. Removing mucilage from Artemisia (*Artemisia sphaerocephala*) seeds also slows germination [17]. Mucilage-producing seeds may improve contact and adhesion to soil particles [8]. A proposed function of mucilaginous seed of *Salvia columbariae* (Lamiaceae) is to collect sand so that the seed is not eaten [5]. Basil-seed mucilage may similarly affect ecology.

ESEM images showed a smooth outer pericarp surface with shallow bumps uniformly spread over the surface where the mucilage was likely stored until hydration (Figure 5A). Hydrochloric-acid treatment removed the mucilage-producing layer. Acid treatment flattened the surface compared to untreated mucilage-producing seeds (Figure 5B). Surface bumps were less evident and surface tears were present following acid treatment (Figure 5B). Dead seed that produced no mucilage had a similar smooth surface appearance to seeds that were acid treated (not shown). The reasons why some seeds in the commercial seed lot were dead is unknown. Dead seeds appeared somewhat smaller, suggesting they were harvested when immature, from dead or unhealthy plants before mucilage formed. Some dead seeds produced small amounts of mucilage. Seeds that were microwaved prior to imbibition

did not germinate but produced as much or more mucilage as fully viable live seeds (not shown). Therefore, mucilage is not required for germination, as seeds with mucilage removed germinated normally (Table 1), and embryos killed by microwaving also produced mucilage. Attempts to record ESEM images during mucilage formation at 100% RH in the sample chamber were unsuccessful because moisture interfered with the electron beam.

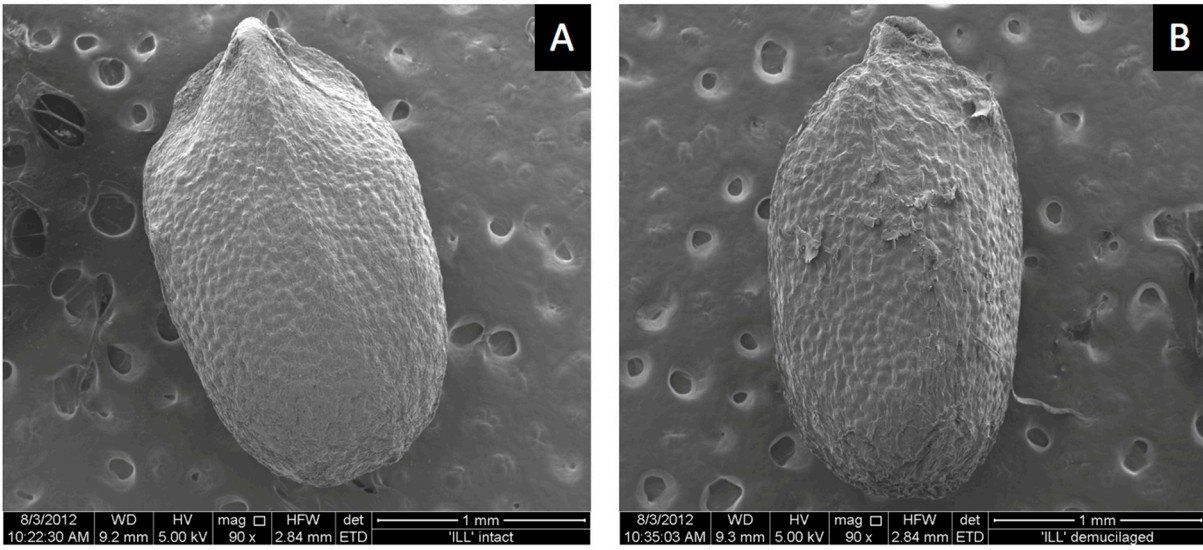

**Figure 5.** Dry intact basil-seed images of sputter coating prior to viewing by ESEM. Mucilage-producing seed (**A**), seed with mucilage removed (**B**).

The hydration (Figure 6) and dehydration (Figure 7) time courses for seeds with and without the mucilage showed distinctive phases. The hydration of both basil cultivars produced similar bimodal seed-imbibition curves with two distinct plateaus. The first increase occurred during the initial 15 min. and corresponded with the hydration of the mucilage layer of the pericarp (Figure 6). The water content increased at a slower rate during the next 3 h until seeds were fully hydrated, when the water content reached a second plateau. Seeds without mucilage produced a single-plateau hydration curve for both cultivars (Figure 6).

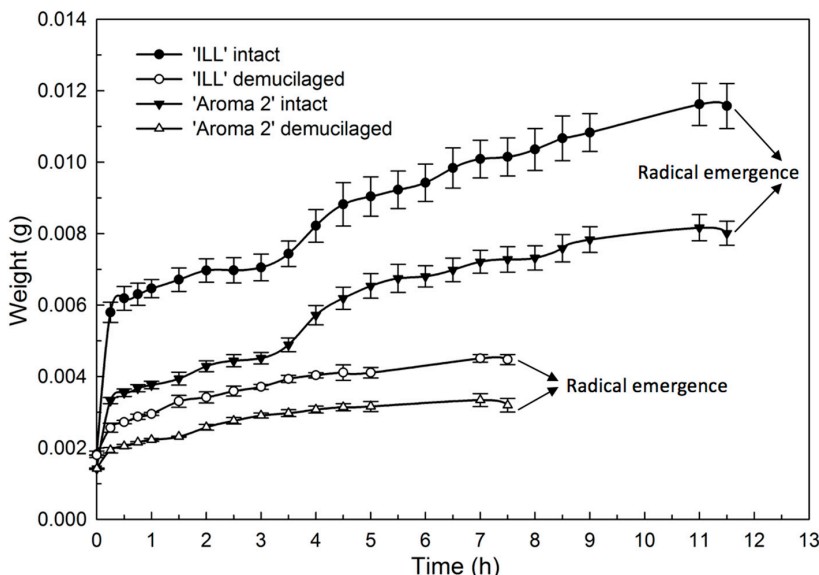

**Figure 6.** Effect of seed-coat-mucilage removal on weight of 'Aroma 2' and 'Italian Large Leaf' sweet basil during seed hydration at 22 °C.

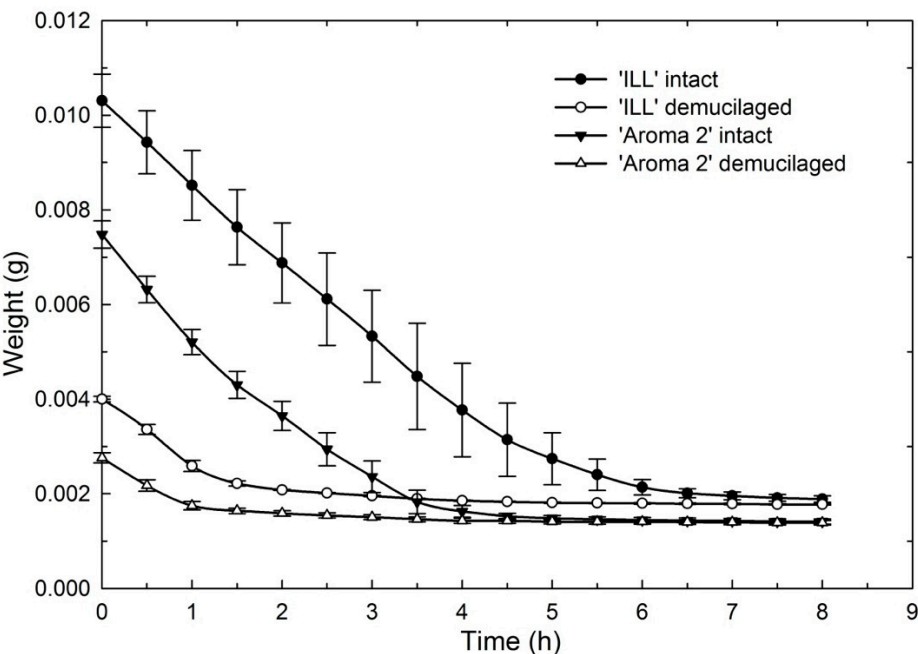

**Figure 7.** Effect of seed-coat-mucilage removal on weight of 'Aroma 2' and 'Italian Large Leaf' sweet-basil cultivars during seed dehydration at 22 °C.

The dehydration curves illustrate that mucilage acts as a water reservoir to maintain hydration for several hours compared to seeds without mucilage that dried more quickly (Figure 7). This would give seeds in nature a significant advantage following rainfall. Rain, even of relatively short duration, could fully hydrate the mucilage, supplying seeds with reserve water to sustain germination and early seedling growth.

The water content of intact seeds for both 'Aroma 2' and 'Italian Large Leaf' was almost four times greater than seeds without mucilage (Table 2). The water-potential measurements of mucilage using osmometry approached 0 MPa, suggesting that the water was loosely bound and readily available for germinating seedlings (Table 3).

**Table 2.** Effect of seed-coat-mucilage removal by acid hydrolysis on water content (dry-weight basis) of 'Aroma 2' and 'Italian Large Leaf' sweet basil hydrated at 22 °C on germination blotter paper for 1 h. Seeds were dried in an oven for 17 h at 103 °C and cooled in a desiccator at 22 °C before weighing. Values are means ± SE of 25 seeds.

| Water Content (Dry Weight) | Intact | Demucilage |
|---|---|---|
| 'Italian Large Leaf' | 4.71 ± 0.43 | 1.28 ± 0.03 |
| 'Aroma 2' | 4.35 ± 0.03 | 1.01 ± 0.02 |

**Table 3.** Water potential of 'Italian Large Leaf' and 'Aroma 2' basil-seed mucilage during hydration at 25 °C from 20 to 150 min. after the start of imbibition measured by osmometry. Each value is the average of two measurements ± standard error.

| Cultivar | Imbibition Time (min) | | | |
|---|---|---|---|---|
| | 20 | 40 | 120 | 150 |
| | | $\Psi$ (MPa) | | |
| 'Italian Large Leaf' | −0.076 ± 0.004 | −0.088 ± 0.003 | −0.106 ± 0.004 | −0.155 ± 0.003 |
| 'Aroma 2' | −0.080 ± 0.004 | −0.094 ± 0.003 | −0.104 ± 0.004 | −0.155 ± 0.003 |

When seeds were hydrated in osmotic solutions of −0.50 and −0.25 MPa, mucilage formation was inhibited at the lower water potential (Figure 8). This illustrates that the water-potential threshold for mucilage formation was above −0.50 MPa. Mucilage only forms when seeds are at high water potentials, which is consistent with the hypothesis that mucilage anchors the seed in the environment after significant rainfall events. This experiment also suggests that the water potential of the mucilage is approximately −0.25 MPa because partial, but not complete, mucilage production occurred (Figure 8). This observation agrees with the direct measurements of mucilage water potential of nearly −0.2 MPa by osmometry after 150 min. for fully imbibed seeds (Table 3). The accuracy of water-potential measurements decreases as one approaches 0 MPa because of the limitations of the isopiestic psychometric technique. Suffice to say, mucilage formation is only possible in well-hydrated environments when seeds approach 0 MPa. Consistent with other studies, imbibition in osmotic solutions slowed the rate of basil-seed germination and mucilage formation (Figure 8).

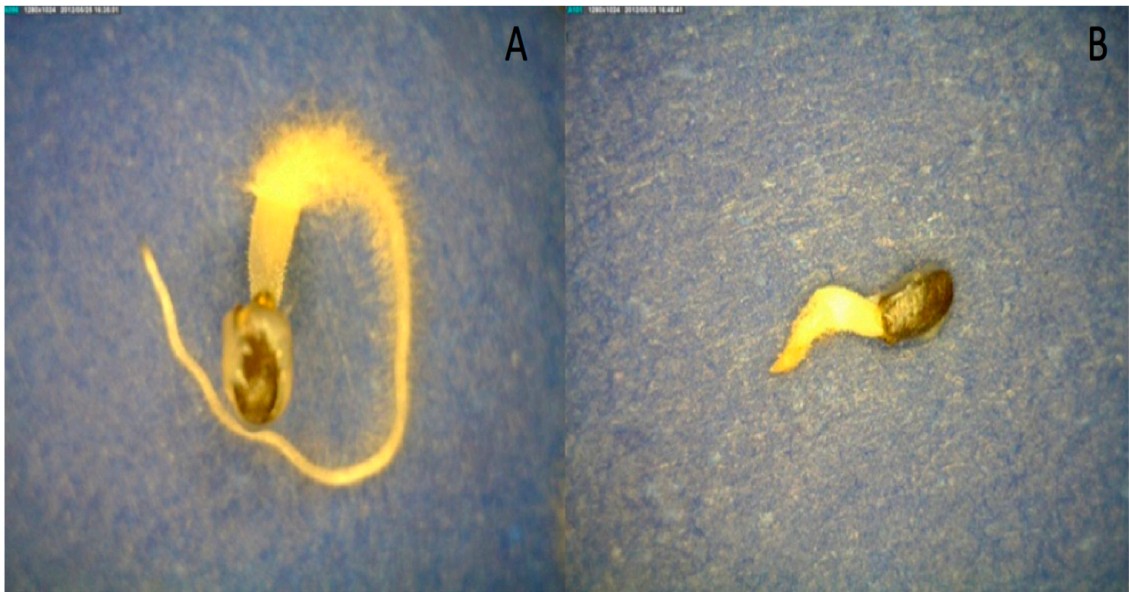

**Figure 8.** Basil germination after four days of imbibition in PEG solutions ((**A**) PEG −0.25 MPa, 25 °C; (**B**) PEG −0.5 MPa, 25 °C).

Hydrated seeds remained anchored by the mucilage to a glass incline board even at a 90° slope for 24 h. Seeds without mucilage had much less adhesion and slid down the glass pane at a 20° slope after 15 min. (not shown). This experiment illustrates that basil mucilage can anchor seeds even on smooth surfaces. This trait may keep basil seeds from moving in a wet environment, anchoring them in moist locations more favorable for germination and establishment.

In an emergence test using potting media, seedling-emergence percentages from seeds with mucilage were higher than the demucilaged seeds when tray cells were initially watered with 4 mL for both genotypes (Table 4). The survival percentage of seedlings after 10 d was higher for seeds with mucilage. Seeds initially watered with 4 mL water showed 22 to 24% higher survival, for 'Aroma 2' and 'Italian Large Leaf', respectively (Table 4). Watering with 6 mL provided sufficient moisture, so no difference in emergence or survival was observed (Table 4). This suggests that mucilage provided a reservoir of water that seedlings could draw upon to survive the initial stages of seedling growth when water was limited.

**Table 4.** Germination and survival percentages of basil seeds (intact and demucilaged) planted in cell trays filled with Sunshine Mix I growth media after 10 days at 25 °C (24 replications of 5 seeds each). Five seeds were placed on top of the media in each cell and 2, 4, or 6 mL of water were pipetted into each cell. Germination was measured by radicle emergence. Surviving seedlings had green cotyledons and white roots after 10 days. Values are means ± SE.

| Genotype/Treatment | Seedling Emergence (%) | | | Seedling Survival (%) | | |
|---|---|---|---|---|---|---|
| | water applied (mL) | | | | | |
| | 2 | 4 | 6 | 2 | 4 | 6 |
| 'Aroma 2' | | | | | | |
| intact | 38 ± 10 | 80 ± 8 | 78 ± 10 | 26 ± 18 | 68 ± 6 | 48 ± 10 |
| *w/o* mucilage | 26 ± 8 | 52 ± 12 | 78 ± 10 | 16 ± 10 | 46 ± 10 | 56 ± 12 |
| 'Italian Large Leaf' | | | | | | |
| intact | 62 ± 14 | 96 ± 6 | 92 ± 6 | 30 ± 8 | 92 ± 6 | 90 ± 6 |
| *w/o* mucilage | 44 ± 20 | 78 ± 8 | 88 ± 8 | 36 ± 10 | 68 ± 8 | 76 ± 10 |

## 4. Conclusions

Hydrophilic mucilage rapidly expanded during the first 20 min. of imbibition from the outer layer of the fruit pericarp surrounding the embryo. The mucilage anchors seeds to a favorable location, preventing movement. Mucilage can be safely removed by dilute hydrochloric-acid treatment prior to planting. The water potential of fully hydrated seeds with mucilage approached zero. Basil seeds with mucilage had almost four times greater water content at full hydration and dried out slower than the those without mucilage. The higher water potential and content demonstrate that mucilage provides a reservoir of loosely bound water. This pool helps seeds germinate and fuels rapid expansive seedling growth aiding establishment in dry environments.

**Supplementary Materials:** The following supporting information can be downloaded at: https://www.mdpi.com/article/10.3390/horticulturae8040327/s1, Video S1: Basil seed hydration.

**Author Contributions:** D.Z. conducted research and received a MS Degree in Horticulture for the work described. G.E.W. was the primary advisor for D.Z., designed experiments, edited the manuscript, measured osmotic pressure of mucilage, and prepared Figure 8. J.N.B. was a graduate committee member of D.Z. who critiqued and edit the manuscript. All authors have read and agreed to the published version of the manuscript.

**Funding:** This research was funded by USDA Regional Research Project W-3168. The Canada Foundation for Innovation, Natural Sciences and Engineering Research Council of Canada, the University of Saskatchewan, the Government of Saskatchewan, Western Economic Diversification Canada, the National Research Council Canada, and the Canadian Institutes of Health Research.

**Acknowledgments:** FTIR qualitative analysis was performed at the Canadian Light Source (CLS). We thank Jarvis Stobbs and Chithra Karunakaran of CLS for FTIR setup and analysis.

**Conflicts of Interest:** The authors declare no conflict of interest.

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
