# Peer review of "Production, Composition, and Ecological Function of Sweet-Basil-Seed Mucilage during Hydration"

_horticulturae, doi:10.3390/horticulturae8040327_

Round 1

Reviewer 1 Report

The background is well written and with sufficient information. The methods are well described, but the statistical analysis should be improved. Two-way ANOVA could be applied to analyze Germination (%) and MTG (days), as well as a more rigorous analysis of Seed Hydration Time Courses and Seed Dehydration Time Courses. I believe that the results should be better integrated, so that it is clear how all the parameters are related. Conclusions fall short due to the lack of analysis. In the end, the FTIR qualitative analysis is not appreciable.

Author Response

We thank you for your helpful comments. 

 Reviewer Comment: Two-way ANOVA could be applied to analyze Germination (%) and MTG (days), as well as a more rigorous analysis of Seed Hydration Time Courses and Seed Dehydration Time Courses.

Author Response: Seed data sets usually violate to assumptions of ANOVA. Germination time data is usually positively skewed and not normally distributed facilitating a log transformation.  Also percentage data are not normally distributed requiring an Arcsin transformation of data prior to ANOVA.  There is an emerging consensus that one should not analyze percentage data with ANOVA.  The point of Table 1 is to show removal of mucilage doesn't change germination percentage or the speed of germination consistently across genotypes. There is no purpose in comparing data across genotypes since they are genetically different and that was not an objective of the study. The percentage data are almost identical for all three cultivars and different statistical methods will not change this.  A paired T test is appropriate because it allows direct pairwise comparison of seeds with and without mucilage which is of primary concern. There is no overall statistical difference among cultivars for speed of germination so mean separation techniques are not appropriate here.  There is interest in differences in pairings of seeds without and without mucilage for each cultivar and that is why the SE values are provided to assess those differences. The time course data are not independent variables so again ANOVA would not be appropriate.  We could model the curves but since the hydration/dehydration time courses are depending on several factors such as media, seed volume, relative humidity, and temperature, such models would not be of benefit to others.  Differences between hydration/dehydration time courses are clear cut and showing SE for each mean we feel is sufficient to support the claims we are making.  Our statistical methods have been approved by a statistician at the Student Statistics Consultation group at Virginia Tech.

Reviewer Comment: I believe that the results should be better integrated, so that it is clear how all the parameters are related. 

Author reply: Yes I agree.  It is unfortunate that Dongfang choose to combine Results and Discussion which was a permitted format for an MS thesis at our University.  This format makes it more difficult for a reader to find and read data of interest. I have tried to change the order to make the progression of results clearer and easier to find.  I have added more figure and table citations to the text.  I think this comment has been addressed in our rewrite.

Reviewer Comment: Conclusions fall short due to the lack of analysis. In the end, the FTIR qualitative analysis is not appreciable.

Author reply: Thank you.  The Conclusions section has been completely rewritten. The FTIR is not quantitative only qualitative but the advantages are that it is very easy to prepare samples for FTIR analysis, only small quantities are needed, it is nondestructive, you can analyze for multiple compounds (e.g. cellulose, pectin, polysaccharides) in a single run that would require different columns, purifications, and runs for HPLC analysis.   For these reasons it was a good choice for validating composition of mucilage in this study and will have utility for other scientists.  We feel this needs to be reported.  We have checked spelling and made corrections as well.  

Reviewer 2 Report

The manuscript by Zhou et al. "Production, composition, and ecological function of sweet basil  seed mucilage during hydration" is well done work. It is well written. The introduction part is ok - it is a good background for the research. The Methods are properly described. The novelty of the work is low, as well as scientific value, however it is a research that may be interesting for the small group of the researchers.

I have some minor comments that may improve the quality of the manuscript.  

Description of the M&M:

the czapter: "Mucilage visualisation" is more like laboratory protocol. In some part may be shortened.

The chapter: 2.8. Germination of Seeds in Media with Different Amounts of Water Added - the title sounds strange. Is it really germination? It is emergence. Germination is finished when you see seed coat protrusion. Please modify the subtitle.

 Results

 Fig. 1. Is it the best of the photos? It would be nice to see the whole seed, not a part of the seed, why is it cut?

 description to fig. 2. Mucilage excretion increasing the total volume of hydrated basil seeds by 5-fold 207 compared to dry seeds (Fig. 2A, B). I can not agree, after analysis of the Fig. 2.

 line 261: Basil is similar to other mucilage producing seeds such as Arabidopsis (what do you mean by similar?)

 Table 1. Could you please explain differences in the number of seeds used in the experiement? 4x25 seeds = 100 seeds (intact), and 3x50 =150 seeds (seeds with removed mucilage)

 Table 2. Could you please explain the unit of water content? It is unclear.

 Table 4 - my suggestion is not to use the word germination but emergence or growth of the seedlings. What do you mean by survival? I understand that it is formation of well developed seedlings. Could you please explain te low survival of intacts seeds in watered conditions (6 ml) - only 48 % in comparison to 68% (seeds watered with 4 ml)?

 Conclusion: Your data do not allow to state "Our results suggest that treatments that mimic natural mucilage production can be  designed and applied to seeds of other species to improve their germination and seedling  survival in moisture-limited conditions". You have shown no data on mimic of natural mucilage production, please modify the sentence.

Author Response

The authors would like to thank the reviews for their helpful comments. 

Reviewer comment (Author response in italics): Description of the M&M:

the chapter: "Mucilage visualisation" is more like laboratory protocol. In some part may be shortened. (The materials and methods have been revised and shortened where possible)

The chapter: 2.8. Germination of Seeds in Media with Different Amounts of Water Added - the title sounds strange. Is it really germination? It is emergence. Germination is finished when you see seed coat protrusion. Please modify the subtitle. (The data in Table 4 is describing seedling growth and seedling survival.  In other experiments, radicle emergence was used as the criteria for germination.  This is a good point and I have clarified Table 4 data by using seedling emergence and seedling survival to avoid confusion with the radicle emergence data presented earlier.)

 Results

 Fig. 1. Is it the best of the photos? It would be nice to see the whole seed, not a part of the seed, why is it cut? (the photo I believe is a 4 micro thin cross section of a basil structure (fruit/seed) to show fusion of the fruit pericarp with the seed.  Thicker sections are challenging to keep the whole field in focus so that is why such a thin section was made.  Of course then the sample is so thin there tends to be separation of the tissues.  We felt this was the best image that showed all the tissues of interest in focus with differential staining) 

 description to fig. 2. Mucilage excretion increasing the total volume of hydrated basil seeds by 5-fold 207 compared to dry seeds (Fig. 2A, B). I can not agree, after analysis of the Fig. 2. (our volume calculation was made based on a three dimensions not just the two shown in Fig. 2)

 line 261: Basil is similar to other mucilage producing seeds such as Arabidopsis (what do you mean by similar?) It is a glucose polymer based with similar viscosity and scaffold structure

 Table 1. Could you please explain differences in the number of seeds used in the experiement? 4x25 seeds = 100 seeds (intact), and 3x50 =150 seeds (seeds with removed mucilage) (We did not compare the same numbers of seeds in these experiments that were conducted at different times.) 

 Table 2. Could you please explain the unit of water content? (It is unclear. water content based on a dry weight basis so it is a percentage)

 Table 4 - my suggestion is not to use the word germination but emergence or growth of the seedlings. What do you mean by survival? I understand that it is formation of well developed seedlings. Could you please explain te low survival of intacts seeds in watered conditions (6 ml) - only 48 % in comparison to 68% (seeds watered with 4 ml)? This has been changed.  I agree this table was confusing and it may have been displayed incorrectly in the draft that you reviewed. 

 Conclusion: Your data do not allow to state "Our results suggest that treatments that mimic natural mucilage production can be  designed and applied to seeds of other species to improve their germination and seedling  survival in moisture-limited conditions". You have shown no data on mimic of natural mucilage production, please modify the sentence.  The conclusions have been rewritten and improved.  Thanks. 

Reviewer 3 Report

The scientific and practical level of the paper is quite high, corresponds to the level of the journal. The text is presented clearly and logically. The structure of the manuscript provides a fairly clear understanding of the process of hydration and mucus formation on the Ocimum basilicum L. seeds when planning subsequent sowing or priming.

For a more complete demonstration of all the advantages of the manuscript, it is desirable to supplement the Introduction section with information on the practical application of the research results, for example, to increase the efficiency of the process of obtaining an improved basil harvest, at the end of the Discussion – to justify and outline future research (for example, the study of the hydration process in empty and fossilized non-viable basil seeds).

I think that the manuscript can be considered for publication in the journal with minor changes (below):

Abstract

  1. Specify the numeric value of high water potential
  2. According to the authors, due to mucus, survival in adverse conditions increases. However, according to their own statement, mucus interferes with automated seeding due, apparently, to adhesive processes.
  3. It is desirable to include the breed of the plant in the keywords, as well as specific parameters measured in the manuscript.

Introduction

  1. The most up-to-date research on this topic dates back to 2017. Are there more recent studies? Please explain.
  2. Clearly and clearly indicate the purpose of the work.

M&M

  1. L 79-80 It is advisable to check the germination of seeds directly during research. If seeds were taken that were stored in 2016 and previously had a germination rate of 85%, it is necessary to explain the storage process and the preparation process for the study.
  2. L 79-80 At 85% germination, at least 15 percent of the seeds will not germinate. However, some of them may be viable. How were empty and fossilized seeds eliminated from the general sample for the representativeness of the final result of full-fledged seeds for hydration?
  3. Specify the methods and criteria for statistical processing in subsection 2.9.

Conclusions:

  1. It is desirable to bring the conclusions in clear agreement with the purpose that is currently missing in the text. It would also be interesting to see here the numerical values of the central trend (mean or median plus or minus the standard deviation) for all the measured parameters - germination, survival, water potential, etc.
  2. The phrase "Our results suggest that treatments that mimic natural mucilage production can be 417 designed and applied to seeds of other species to improve their germination and seedling 418 survival in moisture-limited conditions" cannot be confirmed without conducting studies on seeds of other species (which seeds can form mucus?). I think it is better to use this maxim not in the Conclusions section, but at the end of the Discussion for planning future research.

Author Response

Specify the numeric value of high water potential (approaching 0 MPa, the water potential of liquid water and the limits of psychrometric measurement)

According to the authors, due to mucus, survival in adverse conditions increases. However, according to their own statement, mucus interferes with automated seeding due, apparently, to adhesive processes. Yes this is true.  From an ecological standpoint, mucilage will help seeds survive in their native environment in the Middle East.  For agricultural applications, planting mucilage can be a problem when humidity is high because it can make seeds sticky.  

It is desirable to include the breed of the plant in the keywords, as well as specific parameters measured in the manuscript. I was taught to avoid using the same key words that appear in the title because it is duplicative title words and key words are indexed together by at least some publishers. 

Introduction

The most up-to-date research on this topic dates back to 2017. Are there more recent studies? Please explain.  We added a reference for a review article on this subject from 2021.

Clearly and clearly indicate the purpose of the work. This was clarified

M&M

L 79-80 It is advisable to check the germination of seeds directly during research. If seeds were taken that were stored in 2016 and previously had a germination rate of 85%, it is necessary to explain the storage process and the preparation process for the study. see Table 1 those are our reference germination values and they are in close agreement with those on the packet at the time of sale.  

L 79-80 At 85% germination, at least 15 percent of the seeds will not germinate. However, some of them may be viable. How were empty and fossilized seeds eliminated from the general sample for the representativeness of the final result of full-fledged seeds for hydration?  There is no reported dormancy, primary or secondary, for commercial basil seed.  Only one of the cultivars germinated below 90%.  These percentage are in accordance with those specified in the USA federal seed act which establishes germination standards for seeds sold in the US. 

Specify the methods and criteria for statistical processing in subsection 2.9.

Conclusions:

It is desirable to bring the conclusions in clear agreement with the purpose that is currently missing in the text. It would also be interesting to see here the numerical values of the central trend (mean or median plus or minus the standard deviation) for all the measured parameters - germination, survival, water potential, etc.

The phrase "Our results suggest that treatments that mimic natural mucilage production can be 417 designed and applied to seeds of other species to improve their germination and seedling 418 survival in moisture-limited conditions" cannot be confirmed without conducting studies on seeds of other species (which seeds can form mucus?). I think it is better to use this maxim not in the Conclusions section, but at the end of the Discussion for planning future research.

The Conclusions have been completely rewritten to address these comments.